# Examining Stromal Cell Interactions in an In Vitro Blood–Brain Barrier Model with Human Umbilical Vein Endothelial Cells

**DOI:** 10.3390/cells14110759

**Published:** 2025-05-22

**Authors:** Andrea Margari, Simon Konig, Vignesh Jayarajan, Silvia Rizzato, Giuseppe Maruccio, Emad Moeendarbary

**Affiliations:** 1Department of Mechanical Engineering, University College London, London WC1E 6BT, UK; andrea.margari94@gmail.com (A.M.); simon.konig.21@ucl.ac.uk (S.K.); vignesh.jayarajan.10@ucl.ac.uk (V.J.); 2Omnics Research Group, Department of Mathematics and Physics, University of Salento, CNR-Institute of Nanotechnology, INFN Sezione di Lecce, Via per Monteroni, 73100 Lecce, Italy; silvia.rizzato@unisalento.it (S.R.); giuseppe.maruccio@unisalento.it (G.M.); 3BioRecode Ltd., 75, King William Street, London EC4N 7BE, UK

**Keywords:** blood–brain barrier (BBB), in vitro model, human umbilical vein endothelial cells (HUVECs), endothelial cell biology

## Abstract

Understanding the function of the blood–brain barrier (BBB) in health and disease, as well as improving drug delivery across the BBB, remains a critical priority in neuroscience research. However, current in vitro models of the BBB have become increasingly complex and challenging to implement. In this study, we present a simplified microfluidic BBB model in which human umbilical vein endothelial cells (HUVECs) are cultured as a monolayer along a fibrin gel containing human pericytes and astrocytes. Remarkably, within just three days, the 3D co-culture significantly enhanced barrier formation and upregulated the expression of tight-junction proteins in HUVECs. These findings demonstrate that HUVECs, which have been extensively used for over 50 years to study vascular endothelium due to their ease of isolation and culture, can adapt their phenotype towards that of BBB endothelial cells under appropriate conditions. This microfluidic BBB model offers a valuable tool for drug development and for advancing our understanding of BBB physiology in both health and disease contexts.

## 1. Introduction

The blood–brain barrier (BBB) protects the nervous system by acting as a physical, metabolic, and immunological barrier by selectively regulating the passage of substances from the peripheral blood to the brain [1]. The BBB is composed primarily of brain microvascular endothelial cells, pericytes, and astrocytes, which in turn are surrounded by a vascular basement membrane [1]. This natural dynamic barrier relies on a complex system of receptors, transporters, efflux pumps, and tight junctions to control the entry and expulsion of molecules [2,3]. However, while the BBB plays a critical protective role, it also restricts more than 98% of small-molecule drugs and all macromolecular therapies from accessing the brain [4]. This presents a major challenge for pharmacological interventions in central nervous system disorders, often resulting in insufficient drug delivery to the brain, which in turn leads to low therapeutic efficacy and an aggravation of side effects due to the accumulation of drugs in other organs and tissues [5]. Furthermore, growing evidence indicates that BBB dysfunction plays an important role in a variety of neurological disorders [6,7,8,9]. Thus, understanding BBB function and its disruption is crucial for developing novel therapeutic agents for CNS disorders.

Traditionally, in vivo models have been used to study the role of the BBB in neurological disorders and CNS drug delivery. However, studying the BBB in animal models has various limitations such as interspecies differences in brain physiology that lead to a lack of ability to predict human response, poor clinical translation, and ethical concerns that require continued efforts to minimize the number of animal tests [10,11]. For many years, traditional two-dimensional (2D) in vitro static models, such as transwell assays, have been a key tool in studying the BBB [12]. While such systems are reproducible and easy to use, they only offer a limited representation of the BBB as they lack the correct physiological scale, hemodynamic shear stress, and intercellular interactions: characteristics that play a crucial role in promoting and maintaining EC differentiation into a specific BBB phenotype [13,14,15,16]. For all these reasons, dynamic three-dimensional (3D) BBB in vitro models have been developed. These new models together with the development of materials science and nanotechnology have made it possible to study and test various strategies for regulating BBB permeability, as well as a library of brain-targeted drug delivery systems [5]. Although these models offer a more precise representation of the human BBB’s structure and function, their intricacy results in lengthy and labor-demanding processes.

In this study, we developed an in vitro microfluidic model of the blood–brain barrier that incorporates human umbilical vein endothelial cells as well as human astrocytes and pericytes. This model requires only 3 days of culture, is established in a perfusable microfluidic device, and encompasses 3D juxtracrine interactions between a HUVEC monolayer and astrocytes and pericytes. Our results demonstrate that co-culturing HUVECs with astrocytes and pericytes reduces the permeability of the endothelial monolayer, which is further accompanied by an upregulation of tight-junction proteins.

## 2. Materials and Methods

### 2.1. Protocol Overview

The here described protocol consists of the following central steps: (1) microfluidic device fabrication using soft lithography techniques; (2) expansion of HUVECs, astrocytes, and pericytes; (3) seeding of astrocytes and pericytes in a fibrin hydrogel within the microfluidic device, followed by seeding the adjacent HUVECs to form a monolayer; and (4) culturing of the device for 3 days with daily medium changes. After 3 days, the device is ready to evaluate barrier integrity, assess drug permeability, or perform other readouts.

### 2.2. Model Design

To support the 3D co-culture of HUVECs, astrocytes, and pericytes, microfluidic devices consisting of three adjacent microchannels separated by a series of pillars were fabricated (see Appendix A Figure A1). The central microchannel contained the 3D culture of primary human astrocytes and pericytes within a fibrin gel (“extravascular compartment”) (Figure 1B). On one side of this fibrin gel, a monolayer of HUVECs was subsequently seeded (“vascular compartment”), which created a sealed barrier while allowing for three-dimensional juxtacrine interactions with the extravascular compartment. The two outer microchannels were used to provide the culture medium.

### 2.3. Microfluidic Device Fabrication

Microfluidic devices, made from polydimethylsiloxane (PDMS, Sylgard 184 Silicone Elastomer Kit Dow, Dow, Cheadle, UK), were fabricated with soft lithography techniques. For this, an acrylic mold with a thickness of 400 µm was cut with a laser cutter following a design developed with AutoCAD (version 2024.1.7, Autodesk, San Francisco, CA, USA) (see Appendix A Figure A1). The acrylic mold was glued on an acrylic base. The PDMS (Sylgard 184 Silicone Elastomer Kit Dow, Dow, Cheadle, UK) was prepared by mixing the elastomeric base with the crosslinking agent in a ratio of 10:1, as recommended by the manufacturer. The mixture was then subjected to a degassing process for 40 min inside a desiccator to remove air bubbles. Once degassed, the PDMS was poured into the acrylic molds at a thickness of 5 mm and then polymerized in an oven at 80 °C for one hour. After polymerization, the PDMS replicas were removed from the molds and inlet, and outlet ports for the flow of culture media and hydrogel were created using a 2 mm diameter biopsy puncher. The PDMS replicas were then immersed in deionized water and sterilized by autoclaving. After drying at 80 °C overnight, the PDMS replicas were bonded to #1 glass coverslips with a Corona plasma treater (Elveflow, Paris, France).

### 2.4. Cell Culture

GFP-transduced and non-transduced human umbilical vein endothelial cells (HUVEC, 2B Scientific, Kidlington, UK) were cultured in Endothelial Growth Medium (EGM-2MV, Lonza, Basel, Switzerland) on collagen-coated flasks and maintained in a humidified incubator (37 °C, 5% CO_2_). HUVECs were passaged between 5 and 6 times before experiments. Human brain vascular pericytes (ScienCell, Carlsbad, CA, USA) and human astrocytes (ScienCell) were cultured in the manufacturer’s recommended growth medium (ScienCell) on a poly-l-lysine-coated flask (Sigma-Aldrich, St. Louis, MO, USA) and maintained in a humidified incubator (37 °C, 5% CO_2_). Astrocytes and pericytes were passaged no more than 5 times before being applied in experiments.

### 2.5. Lentiviral Transduction

To inspect the location of cells further, we fluorescently labeled astrocytes and pericytes using lentiviral transduction. For this, 1 × 10^5^ astrocytes or pericytes were plated in T25 flasks containing complete culture medium. After 24 h, the culture medium was replaced with Opti-MEM (Gibco, New York, NY, USA), and cells were transduced with either LV-EF1α-mCherry/eGFP (VectorBuilder, Chicago, IL, USA; #LVI(VB010000-9492agg)) or LV-SFFV-mCherry (kindly gifted by Dr. Xin Huang, UCL) (see Appendix A Figure A2) at a multiplicity of infection (MOI) of 5. The transduction was carried out for 8 h, after which the viral supernatant was removed and replaced with fresh complete culture medium. Cells were then cultured for an additional 4 days prior to their use in subsequent experiments.

### 2.6. Three-Dimensional Cell Culture in the Microfluidic Platform

On day 0, astrocytes and pericytes were initially seeded inside the fibrin gel, followed by the seeding of a monolayer of HUVECs in one of the external microchannels. For the preparation of the fibrin gel, fibrinogen (Sigma-Aldrich) was dissolved at a concentration of 6 mg/mL in sterile PBS, while thrombin (Sigma-Aldrich) was initially diluted to 100 U/mL in sterile PBS and then further diluted with EGM-2 MV medium (Lonza) to reach a final concentration of 4 U/mL, to be used in the experiments. Astrocytes and pericytes were detached and centrifuged at 200 rpm for 5 min then resuspended in the thrombin solution. To form the fibrin gel, the fibrinogen solution was mixed 1:1 with the cell suspension containing thrombin and immediately pipetted into the central channel of the microfluidic device.

In the fibrin gel, the final concentrations of astrocytes and pericytes were 2 × 10^6^ cells/mL and 1 × 10^6^ cells/mL, respectively. In experiments where astrocytes or pericytes were not used, fibrinogen was mixed with the thrombin solution without the addition of cells. To allow the gel to polymerize, the devices were placed in a Petri dish and placed in a humidified incubator at 37 °C with 5% CO_2_ for 30 min after seeding. Once the gel polymerization occurred, the external channels were treated with human fibronectin (60 µg/mL) (Thermo Fisher Scientific, Waltham, MA, USA) for 30 min in a 37 °C incubator with 5% CO_2_ to promote endothelial cell adhesion. To form the endothelial monolayer, HUVECs were collected and seeded at a density of 1 × 10^6^ cells/mL in EGM-2M medium (Lonza) in one of the side channels. After seeding the endothelial cells, the devices were positioned inside the incubator at an inclination of approximately 60° for one hour. This inclination allowed the cells to distribute uniformly along the gel wall, thus facilitating the formation of a continuous monolayer of endothelial cells, essential for modeling the BBB inside the device. After this period, the device was kept in an incubator at 37 °C with 5% CO_2_ for 3 days, with medium replacement every 24 h.

### 2.7. Permeability Measurements

After 3 days of culture, the permeability of the HUVEC monolayer was assessed by adding a fluorescent dextran tracer (FITC-dextran at 70 kDa; 10 µg/mL in EGM-2MV; Life Technologies, Carlsbad, CA, USA). Imaging of the dextran, endothelium, and fibrin gel was performed using an inverted epifluorescent microscope (Leica DMi8, Leica, Wetzlar, Germany) (equipped with a motorized stage and live-cell imaging modules) with a 10× objective, acquiring images every 30 s for a period of 20 min. Permeability was then calculated by applying Equation (1), as previously used similarly [17]:(1)$$P=\frac{1}{t_{2}-t_{1}}\frac{I_{gel}^{t_{2}}-I_{gel}^{t_{1}}}{\left[\left(I_{top}^{t_{2}}-I_{gel}^{t_{1}}\right)+\left(I_{top}^{t_{2}}-I_{gel}^{t_{1}}\right)\right]/2}d$$
in which *I_gel_* and *I_top_* are the average intensity of the tracer in the gel and in the media channel next to the HUVEC monolayer, respectively. *t*_1_ and *t*_2_ represent the initial and final time points, and *d* is width of the gel.

### 2.8. RNA Isolation and Real-Time Quantitative Polymerase Chain Reaction (RT-qPCR)

At the end of the experiments, after 3 days in culture, the HUVECs were isolated from devices where either HUVECs, pericytes, and astrocytes had been co-cultured or HUVECs had been cultured alone. Media were completely removed from the media channels, and HUVECs were detached by adding 1X TrypLE (Gibco) for 5 min to the channel where the HUVEC monolayer had formed, incubating for 5 min. Once detached, the TrypLE cell suspension was collected into a microcentrifuge tube. To recover any remaining cells, the channel was rinsed twice with EGM-2M medium, and the rinse was added to the same tube. Devices were inspected to ensure that mainly the HUVEC monolayer had been collected, while most of the astrocytes and pericytes had remained inside the fibrin gel. HUVECs were combined from 5 devices for each experimental condition. Cells were pelleted, followed by lysis and RNA extraction using the RNeasy Plus Mini kit (Qiagen, Germantown, MD, USA). The RNA concentration and purity were assessed with a NanoDrop (DeNovix, Wilmington, DE, USA). RNA was diluted to 7 μg/mL for cDNA synthesis using the High Capacity cDNA Reverse Transcription Kit (Applied Biosystems, Waltham, MA, USA). Quantitative real-time RT-PCR (RT-PCR) using SsoAdvanced Universal SYBR Green Supermix (Bio-Rad Laboratories, Herculais, CA, USA) was performed using the primers provided in Appendix A Table A1. qPCR data were analyzed using the ΔΔC_t_ method, with platelet endothelial cell adhesion molecule-1 (*PECAM-1*) (expressed by HUVECs but not by astrocytes or pericytes) used for normalization to account for non-endothelial RNA, as in prior studies [18].

### 2.9. Immunofluorescent Staining and Image Acquisition and Analysis

Devices were cultured for 3 days, then rinsed with PBS and fixed in 4% paraformaldehyde (PFA, Electron Microscopy Sciences, Hatfield, PA, USA) for 15 min at room temperature (RT). To permeabilize cell membranes, 0.1% Triton X-100 (Sigma-Aldrich) was used in PBS for 20 min at RT and then blocked with 0.1% normal goat serum (Abcam, Cambridge, UK) in 0.1% Triton X-100 solution for 1.5 h at RT. Primary antibodies included anti-CD31 (#561654, BD Biosciences, Franklin Lakes, NJ, USA), anti-glial fibrillary acidic protein (GFAP, #ab33922, Abcam), and anti-zonula occludens-1 (ZO-1, #617300, Invitrogen, Waltham, MA, USA), diluted (1:100) in the blocking solution and applied for 24 h at 4 °C. Devices were washed 4 times for 10 min by adding PBS to the outer channels. After washing, secondary antibodies (Anti-rabbit Alexa Fluor 488, #4412S, Cell Signaling, Danvers, MA, USA) were applied in PBS for 2 h at room temperature. For f-actin visualization, phalloidin probes (165 nmol, #A12381, Invitrogen) were used for 20 min at RT. Nuclei were counterstained with 6-diamidino-2-phenylindole (DAPI, 1:1000, Sigma-Aldrich) for 15 min. Before imaging, samples were rinsed again with PBS 4 times for 10 min at RT. Images were taken using an inverted confocal microscope (Fluoview FV1000, Olympus, Tokyo, Japan).

### 2.10. Statistics

Permeability coefficients were compared by applying a two-tailed two-sample Student’s *t*-test in Excel. The results were considered significant if *p* < 0.05.

## 3. Results

### 3.1. An In Vitro Model of the Human Blood–Brain Barrier Consisting of HUVECs, Astrocytes, and Pericytes

To develop an in vitro model of the blood–brain barrier (BBB), we utilized a three-channel microfluidic device for the 3D co-culture of human umbilical vein endothelial cells (HUVECs), human astrocytes, and brain pericytes. The structure of the brain’s cerebral vasculature was mimicked by culturing a monolayer of HUVECs, referred to as the “vascular compartment”, along the sides of a fibrin-based hydrogel (Figure 1B,C). Within this hydrogel, human astrocytes and brain pericytes were co-cultured to replicate the extravascular environment of the BBB.

Notably, after just three days of co-culture, immunohistochemical analysis of the HUVEC monolayer revealed the presence of not only the typical endothelial marker CD31 at cell–cell junctions but also ZO-1, a key protein involved in tight-junction formation at the BBB (Figure 1D). Additionally, fluorescently labeling of astrocytes and pericytes demonstrated that, within the fibrin hydrogel, both astrocytes and pericytes were establishing direct connections with the HUVEC monolayer (Figure 1E,F).

### 3.2. Impact of Astrocytes and Pericytes on HUVEC Barrier Formation

In the human blood–brain barrier (BBB), stromal cells such as astrocytes and pericytes play a crucial role in supporting endothelial cell barrier formation [1]. To determine whether this also applies to our in vitro BBB model, we cultured HUVECs alone and in combination with astrocytes, pericytes, or both cell types (Figure 2). Barrier integrity was assessed by measuring the permeability to 70 kDa dextran (Figure 2A). As anticipated, our results showed that the permeability coefficient significantly decreased with the addition of astrocytes and pericytes (Figure 2B). Specifically, permeability dropped from an initial value of 1.48 × 10^−4^ cm/s, observed with the HUVEC monolayer alone, to progressively lower values when either astrocytes or pericytes were incorporated into the gel. The greatest reduction in permeability, reaching a minimum of 0.25 × 10^−4^ cm/s, was achieved when both astrocytes and pericytes were co-cultured.

To further explore whether this decrease in permeability was linked to enhanced barrier formation, we assessed the expression of tight-junction proteins Zona Occludens-1 (ZO-1), Claudin-5 (CLDN5), and Occludin (OCLN) via RT-qPCR in both mono- and co-culture conditions (Figure 2C). Interestingly, *CLDN5* and *OCLN* expression was upregulated when HUVECs were co-cultured with astrocytes and pericytes in the microfluidic devices. *ZO-1* expression levels remained unchanged between the two conditions; however, immunocytochemistry revealed enhanced junctional localization of ZO-1 and the formation of more continuous cell–cell junctions (Figure 2D).

## 4. Discussion

In this study, we developed a blood–brain barrier (BBB) model on a chip consisting of a vascular compartment that connects directly to a perivascular compartment containing human primary brain astrocytes and pericytes, thereby closely mimicking the structural organization of the human BBB. The model is distinguished by its quick assembly as well as the opportunity to observe phenotypic changes of HUVECs upon co-culture with brain vasculature stromal cells. Given that we used a microfluidic chip that could readily enable perfusion across the HUVEC monolayer, this model has the potential to serve as a foundational platform for future pharmacological studies assessing drug penetrance across the BBB.

Due to the ease of permeability assessment, the BBB is most commonly mimicked in vitro using the principle of a transwell assay [11]. However, due to its 2D nature as well as the use of a stiff membrane between cells, transwell assays can only model the complex 3D interactions at the BBB in a limited way [12]. Over recent years, increasingly complex 3D models have been developed for permeability analysis including configurations whereby 3D tubular vasculature is assembled either via creating a luminal channel or via de novo vasculogenesis [19,20,21]. While this achieves a high degree of biomimicry, such complex systems require a high level of expertise and are laborious. The model presented here serves as a combination of these approaches whereby it is able to recapitulate complex 3D juxtacrine interactions while also allowing the simple assessment of drug penetrance across a monolayer of BBB endothelial cells.

In agreement with previous studies, we have shown that endothelial cells increased their expression of tight-junction proteins when co-cultured with astrocytes and pericytes [22]. Interestingly, only when pericytes were incorporated within the BBB model, either with or without astrocytes, there was a significant reduction in its permeability compared to HUVECs alone. This is in line with previous studies and indicates the important role of pericytes in regulating BBB permeability [18]. The permeability coefficient was, as expected, the lowest when both astrocytes and pericytes were co-cultured with the HUVECs. It should be noted that the permeability coefficients achieved here are two orders of magnitude higher than those reported for rat brain capillaries as well as for BBB or HUVEC microvasculature models [18,23,24,25]. However, due to the use of HUVECs which are of non-brain origin, such a decrease in barrier tightness is to be expected and aligns with previous studies where HUVECs were cultured as a monolayer across a fibrin or collagen gel [26,27].

It should be noted that HUVECs may not be fully suitable for modeling the human BBB microvasculature, as their origin from large veins results in notable differences in barrier function, gene and protein expression, and interactions with immune and tumor cells compared to the specialized brain microvascular endothelial cells that comprise the BBB [22,28]. Despite this, our results indicate that co-culture with BBB stromal cells induces tight-junction expression, which is an integral part of the barrier-forming properties of the BBB. Previous studies showed that HUVECs express BBB tight-junction proteins when cultured with media containing bovine brain extract as well as during co-culture with bovine pericytes [22,29]. Interestingly, upon transplantation into mice brains, HUVECs form connected microvessels with BBB-like barrier properties, GLUT1 expression, and connections to surrounding astrocytes [30]. Collectively, our results and previous research thus indicate that HUVECs may be able to change their phenotype depending on the surrounding microenvironment. HUVECs have been used for over 50 years to study vascular biology in vitro and can be relatively easily isolated and cultured. Therefore, in vitro BBB models based on HUVECs, such as those presented here, could constitute an alternative to more complex models or models where immortalized cell lines are used.

It is important to note that our model only partly incorporates juxtracrine interactions between endothelial cells, pericytes, and astrocytes. In vivo, pericytes cover approximately 30% of the extraluminal side brain of capillaries with astrocytes forming endfeet, which cover nearly the whole outer surface area of brain capillaries [31]. While direct contact between HUVECs and stromal cells could be observed, increasing cell numbers or longer culture time may have increased more mature interactions.

While the model presented here utilized a microfluidic chip, we did not incorporate fluid flow, which has been shown to be essential for proper blood–brain barrier (BBB) function [15]. However, future studies using this model could readily incorporate pumping systems connected to the device outlets or apply hydrostatic pressure, as demonstrated in similar microfluidic setups, to enable controlled fluid flow along the HUVEC monolayer [32]. By tracking the transport of drugs or drug delivery systems from the media channel into the fibrin gel across the endothelial layer, the presented model could serve as a practical platform for evaluating drug permeability across the BBB. Furthermore, the system may be adapted for future studies of BBB pathophysiology by enabling the controlled introduction of barrier-disrupting agents. It could also be used to investigate how dysfunction in stromal cells, such as pericytes or astrocytes, may influence the barrier properties of the endothelial monolayer.

## Figures and Tables

**Figure 1 cells-14-00759-f001:**
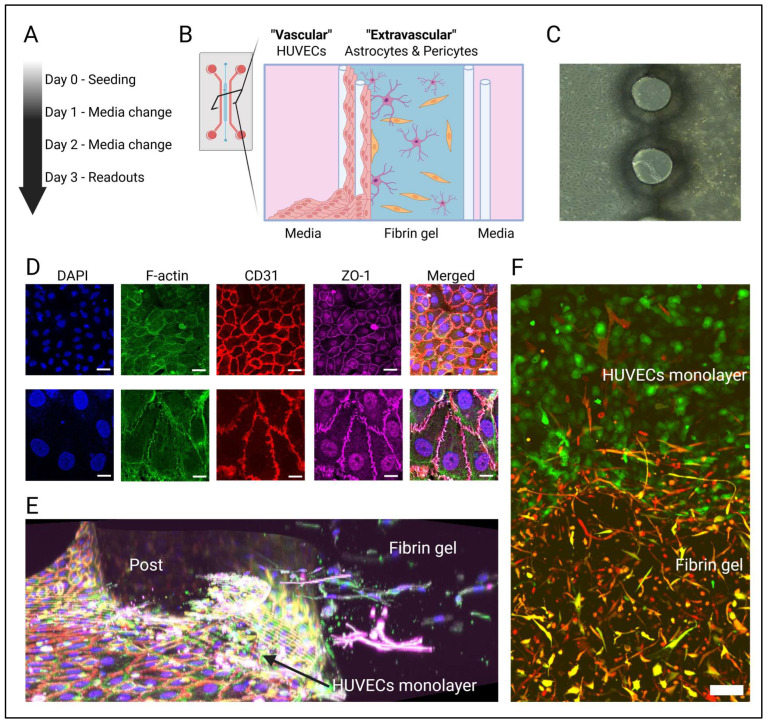
A blood–brain barrier (BBB) model on a chip composed of human umbilical vein endothelial cells (HUVECs) as well as human brain astrocytes and pericytes. (**A**) Methodology and timeline to establish the BBB model. (**B**) Astrocytes and pericytes were cultured inside a fibrin gel (“extravascular compartment”) within the central microchannel of a microfluidic device. Along one of the sides of the fibrin gel, a HUVEC monolayer was seeded (“vascular compartment”) which tightly sealed the fibrin gel while allowing for 3D juxtracrine and paracrine interactions with the extravascular compartment. Created with BioRender.com. (**C**) Brightfield image showing the HUVEC monolayer and cell-loaded fibrin gel separated by an array of microposts. (**D**) Confocal images show expression of BBB-specific marker ZO-1 at cell junctions along F-actin and CD31. Upper panel 20× magnification (Scale bar = 30 µm), lower panel 60× magnification (Scale bar = 10 µm). (**E**) 3D projection of HUVEC monolayer along the fibrin gel. The HUVEC monolayer was stained for f-actin (green) and CD31 (red), while GFAP staining (purple) showed astrocytes within the fibrin gel. Other cells within the fibrin gel were labeled for f-actin (green) but not GFAP and thus revealed as pericytes. Nuclei are stained with DAPI (blue). (**F**) Maximum intensity z-projection of confocal image taken across the device. HUVECs expressed GFP (green), pericytes expressed mCherry (red), and astrocytes expressed both GFP and mCherry (yellow). Scale bar = 100 µm.

**Figure 2 cells-14-00759-f002:**
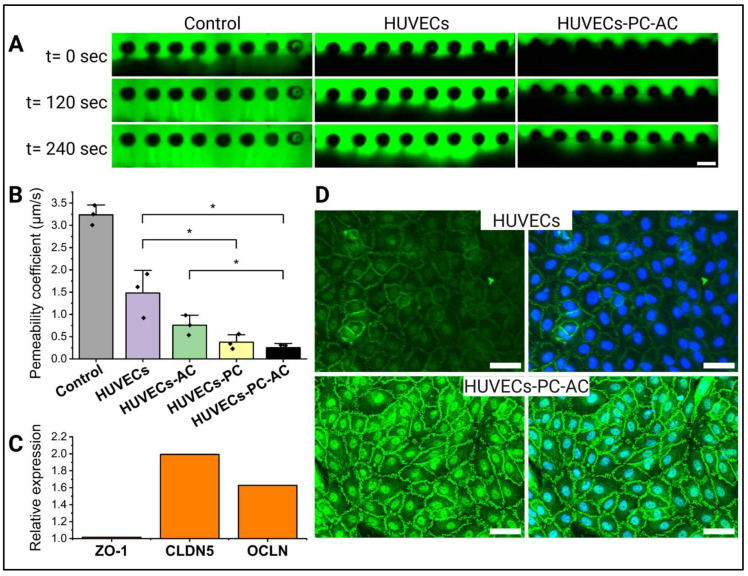
Brain vascular stromal cells affect barrier formation and tight-junction expression by HUVECs. (**A**) 70 kDa dextran (green) was injected into one of the media channels along the HUVEC monolayer to measure barrier integrity. Representative images showing dextran infusing into the gel but at different speeds depending on the experimental conditions. Scale bar = 500 µm. (**B**) Permeability values (µm/s) decreased depending on whether no cells were added (control), a HUVEC monolayer was added (HUVEC), or this monolayer was co-cultured with astrocytes (HUVEC-AC), pericytes (HUVEC-PC), or both (HUVEC-PC-AC). (*n* = 3 devices per experimental condition.) Horizontal lines with asterisks indicate statistically significant differences between groups (*p* < 0.05). (**C**) RT-qPCR analysis of Zona Occludens-1 (*ZO-1*), Claudin-5 (*CLDN5*), and Occludin (*OCLN*) expression in HUVECs isolated from HUVECs-PC-AC co-culture devices versus HUVECs monocultures. (RNA was pooled from 5 devices per experimental condition). (**D**) Immunocytochemistry for ZO-1 revealed increased presence at cell contact points in HUVECs co-cultured with astrocytes and pericytes compared to monoculture conditions. Images were taken of the HUVEC monolayer close to the gel interface. Scale bars = 50 µm.

## Data Availability

Data will be made available on request.

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
