# Peer review of "Examining Stromal Cell Interactions in an In Vitro Blood–Brain Barrier Model with Human Umbilical Vein Endothelial Cells"

_cells, 2025, doi:10.3390/cells14110759_

Round 1

Reviewer 1 Report

Comments and Suggestions for Authors

This work proposes an original model for the ex vivo reconstitution of the blood-brain barrier using HUVEC, which are more readily available than brain endothelial cells. 
After a brief description of the experimental set-up, the results show that the addition of pericytes and astrocytes induces a phenotype similar to that of BBB through expression of the ZO-1 junction protein and the inability of a 70KDa dextran to pass from the inside to the outside of the HUVEC layer.
My first question concerns the device: is it pressurised or is a flow applied to the “luminal” side, i.e. the surface of the cell layer formed by the HUVECs?
This preliminary work could be completed by analysing the expression of other junction proteins (Claudin-5, for example). It is necessary to show that ZO-1 expression does indeed depend on the addition of pericytes and astrocytes to the culture (in the same spirit as figure 2B).
It also important to see pericyte in coculture with the specific labelling of pericytes in cocultures.
ZO-1 labelling (figure 1D) seems to indicate that the nuclei of HUVEC cells can be revealed by the ZO-1 antibody (a figure showing the specificity of secondary antibodies and labelling with another anti-ZO1 would be reassuring).
In several places there are error Reference source not found which suggests that data is missing.

The identifiers of the primary and secondary antibodies are missing.
I think that the AQP4 labelling that the authors refer to as unpublished data should be added to the MS to reinforce the discussion.

Reviewer 2 Report

Comments and Suggestions for Authors

In this study, the authors demonstrate  a microfluidic BBB model in which human umbilical vein endothelial cells (HUVECs) are cultured as a monolayer along a fibrin gel containing human pericytes and astrocytes. While this microfluidic BBB model appears to be a preliminary tool for the BBB study, there are several flaws as below to prevent its publication: 1) BBB consists of microvascular endothelial cells (ECs) but HUVECs are ECs derived from big vessels; 2) there is lack of comparison in the BBB structures between the in vitro cultured 3D model and in vivo tissues; 3) there is lack of vigor to show the utilities or usefulness of the presented in vitro cultured 3D BBB model for studying the functions of physiological nd pathophysilogical conditions.

Round 2

Reviewer 1 Report

Comments and Suggestions for Authors

Dear colleagues,

Thank you for your answers to my questions and for the modifications added in the MS.

Best regards.

Reviewer 2 Report

Comments and Suggestions for Authors

No more.